# DISCERNING HYDROCLIMATIC BEHAVIOR WITH A DEEP CONVOLUTIONAL RESIDUAL REGRESSIVE NEURAL NETWORK

## ABSTRACT

Water impacts the globe daily in new and familiar ways such as the ongoing western United States drought and the 2022 Pakistan flood. These events sustain uncertainty, risk, and loss forces to the global ecosystem. Better forecasting tools are mandatory to calibrate our response in an effort to mitigate such natural hazards in our watersheds and adapt to the planet's dynamic environment. Here, we present a Deep Convolutional Residual Regressive Neural Net (DCRRNN - pronounced "discern") platform for obtaining, visualizing, and analyzing the basin response of watersheds to water cycle fluxes. We examine four very large basins, simulating river response to the hydroclimatic fluxes they face. Experiments modulating the lever of time lag between remotely sensed and ground truth measurements are performed to assess the metrological limits of this forecasting device. The resultant grand mean Nash Sutcliffe and Kling Gupta efficiency values are both of greater value than 90%. Our results show that DCRRNN can become a powerful resource to simulate and forecast the impacts of hydroclimatic events as they relate to watershed response in a globally changing climate.

## 1 INTRODUCTION

Water is connected to and connects all living things on Earth. It is wielded to power electronic devices, enables plants, food and animals to grow, serves as the living and recreational space for many creatures big, small, young and old, and is nourishment to the human body. It has been both the subject of, platform for, and weapon of choice in numerous conflicts. Global hydraulic infrastructure is highly variable. Dirty water can be a source of disease and death. Water is branded, modified, and sold at differing levels of purity and concentration. The cost of equipment to control the flow of water is high, maintenance is frequent, and change of demand and supply is a constant source of concern.

Furthermore, human activities have changed and continue to change Earth's environment. The changes are visible in both short (meteorological) and long (climatological) time scale responses (Stott, 2016). As the temperature of our home planet increases, the amount of snow and sea ice loses volume over time (Qin et al., 2020; Min et al., 2022), sea levels rise and swallow up once inhabited land (Tebaldi et al., 2021; Sévellec et al., 2017), storms intensify (Karl et al., 1997), droughts last longer Underwood (2015), floods become more severe (Milly et al., 2002; Hirabayashi et al., 2013), animal populations go extinct (Parmesan et al., 2000), and the availability of freshwater becomes more unreliable (Gleick & Cooley, 2021).

Concurrently, manmade Earth observation and control systems continue to improve (Crisp et al., 2020; Minzu et al., 2021). Research, operational and pedagogical software tools for the climate sciences are interrelated by common programming interfaces and standards. In these development environments, the handling and organization of data is paramount for usability. In the United States, government supported big data systems warehousing climate data are mature. Here, we approach the topic of watershed modeling with a learned representation. We observe the connections between model output of four United States drainage basins to actual gauged in the river measurements. All basins are greater than 1M acres and one upwards of 1B. Each are substantial in size to observe how the change in runoff and subsurface flow impacts the quantity of water discharging from the

major river within the basin. Our findings show that the neural network performs well on all basins according to commonly referenced statistical metrics when satellite-derived flux of water and ground truth gauged streamflow are captured on the same day. The framework presented here meets or exceeds comparable studies in the basins selected. Given the validated efficiency values, we envision future work applying the same tools to study and consider all of Earth's watersheds at fine fidelity.

## 2 MATERIALS AND METHODS

### 2.1 STUDY AREAS

Four United States drainage basins with areas of greater than one million acres each were selected as study areas and are shown in Figure 1. The Bear River and Connecticut River watersheds are significantly smaller than either the Mississippi River or the Colorado River basins. The model output imagery used observes approximately 100 square kilometers of area (on the order of 25,000 acres) in each pixel. Therefore, sufficiently large basins must be selected to ensure an adequate number of pixels per day per location.

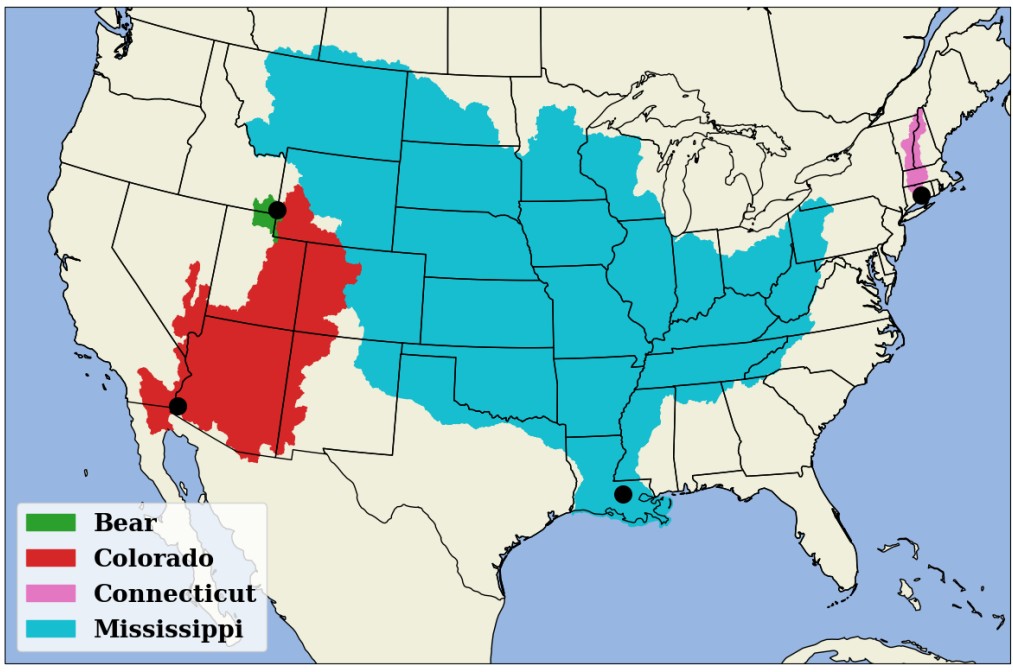

Figure 1: Drainage basins under investigation

### 2.2 SATELLITE-DERIVED MODEL OBSERVATIONS

For each basin there are two input images, otherwise known as channels. These images are extracted from raw data obtained through the NASA Goddard Earth Sciences Data and Information Services Center. The raw data is National Land Data Assimilation System (NLDAS) model output. NLDAS is a project run by several United States based institutions and universities. NLDAS takes continental scale meteorological data parameters (e.g. air temperature, wind speed, surface pressure, precipitation, incoming radiation, specific humidity) as input and deterministically creates water and energy flux layers as outputs. The NLDAS project in its second phase applies several different water and energy balance algorithms to create flux outputs from one common set of meteorological inputs. Here, the Noah implementation of a water and energy budget algorithm is used. Noah is selected because of its concurrent implementation in the Global Land Data Assimilation System and length of time series. The channels of interest are components of water flux, specifically surface and sub-surface runoff, as they collectively represent the lateral movement of liquid water along and under

the surface towards the terminal drainage point at a given point in time (Xia et al., 2012; Liang et al., 1994).

Figure 2: NLDAS daily surface and subsurface flows

## 2.3 Ground Truth Measurements

Concurrent with the two NLDAS channels is a single gauged in the river streamflow measurement. Daily streamflow measurements from four sites near the terminus of each basin are obtained from the United States Geological Survey's National Water Information System. The USGS operates nearly 30,000 daily streamflow data collectors (Edwards et al., 1986). Sites were selected based on the availability, proximity to the terminal point of the basin, and relative continuity of data. Gaps in data collection are solved with linear interpolation.

## 2.4 Data Collection and Preprocessing

For this study, we looked at the time range starting on January 1, 2015 until the most recent output available, March 1, 2022. The NLDAS model output is available in a monthly and hourly product. We elected to combine the hourly data available for surface and subsurface streamflow into a daily product. The raw hourly NLDAS product with all variables is a directory sized 351 gigabytes comprised of 62,805 hourly files. The summing and extraction of lateral flows shrunk the total file size by a factor of more than 150. Each raw data file consumes 5.8 megabytes of disk space, while each daily surface and subsurface flow extraction 822.7 kilobytes. Filtered data consumes only 2.1 gigabytes and can easily be held on a graphical processing unit when trained with the neural network. File size decreases further when clipped to a particular basin. Images are z-scored relative to themselves while gauged streamflow data is z-scored relative to the entire time series of seven years. Whitening has been shown to improve the performance of training a neural network (Karhunen et al., 1997; Chen et al., 2020).

## 2.5 Neural Network Architecture

In this instance, images of Earth's surface and subsurface water flow are passed through the network that has random numbers associated with each layer and node of the network. Eventually, the transformed image values reach a destination where its shape matches that of the target of the input pair; here, the target is one pixel as the daily value for gauged streamflow is a single physical measurement. The problem is one of regression because the prediction of streamflow is continuous and can theoretically be any value greater than zero. We use convolutional neural networks because our input to the network is a sequence of two channel images (Rawat & Wang, 2017). We also use residual learning, which allows us to make the network very deep but control the opacity of the

initial structure of the image. This makes training faster (He et al., 2016). Rectified linear unit activation functions are applied in all but the last layer of nodes, and batch normalization is used in the residual layers (Agarap, 2018; Ioffe & Szegedy, 2015). Batch normalization is similar to the z-score treatment applied in 2.4. We selected a variant of stochastic gradient descent for optimization of the neural network nodes (Amari, 1993; Kingma & Ba, 2014).

## 3 RESULTS

Hourly NLDAS model output of surface and subsurface flow are summed to daily accumulations over the time span of January 1st, 2015 to March 1st, 2022. This time series is 2,617 long comprised of two channel images. Channels are surface and subsurface flow in kilograms per square meter. Units are analogous to the weight of water in a given location. Sample observation output from each basin capturing flow behavior on June 6th, 2021 is displayed in Figure 2. The effects of spatial resolution are apparent, as the Bear River and Connecticut River basins have pronounced rectangular edges due to their relatively small size. This pixelation effect is not present in the Mississippi River and Colorado River observations of lateral flow from the basin view at this constrained figure size.

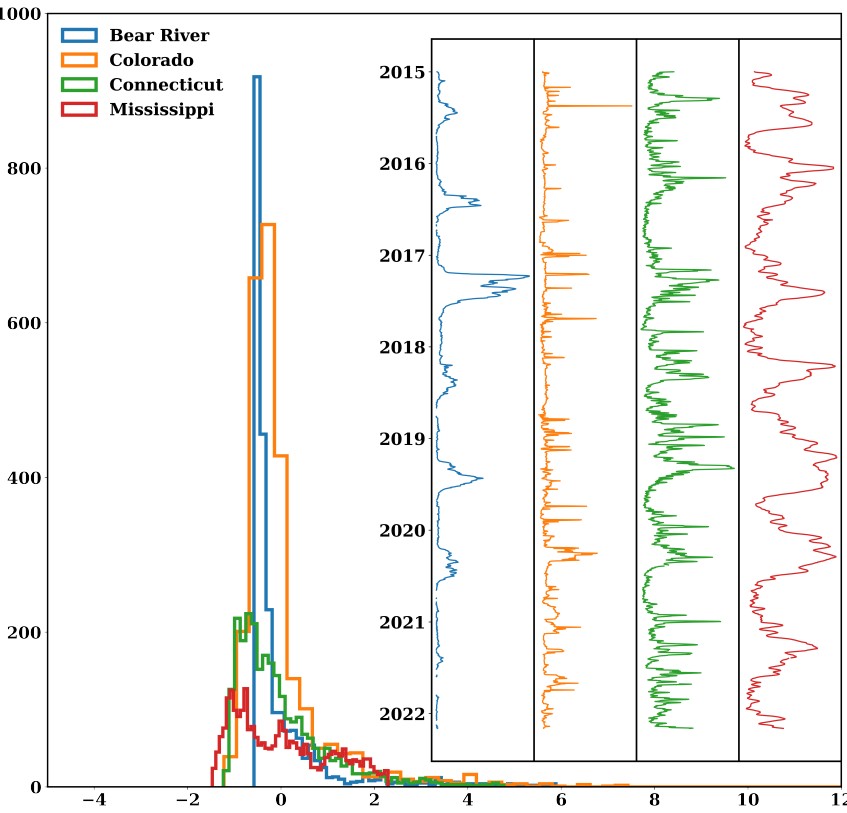

Figure 3: Strip chart and histogram plots of z-scored gauged streamflow observations

Gauged streamflow measurements of the four target rivers are significantly different in magnitude from one another. We process each with a z-score treatment to center their mean values around the number zero and standardize each increasing and decreasing integer around intervals of standard deviation. Measurements of each basin both as a function of time and flow are presented in Figure 3. The strip charts show the change in streamflow over time, and the histograms show how often actual measurements in the respective basin occur relative to the average discharge. This is a single dimensional z-scoring system. We also perform a two dimensional treatment to each of the input channels, surface and subsurface streamflow. Whereas the 1-D treatment uses the entire time series of gauged streamflow measurements for computation, 2-D z-scores are computed based on a single image at a time. Changeable levers to control DCRRNN are basin, lag, number of epochs and the

ratio of training data to testing data (TTS). There is also an override for stopping the model training early when the training data has a Nash Sutcliffe efficiency (NSE) value of a variable efficiency percentage.

Figure 4 shows a sample output from one configuration of the neural network. The topmost graph illustrates the time series of discharge measurements in cubic feet per second of the Bear River. This graph is rotated ninety degrees relative to its sibling hydrograph in Figure 3. There is a notable seasonality to this streamflow measurement of Bear. Spring brings melting snow pack in the nearby mountainous terrain and subsequent increases in neighboring river flows. Spring melting snow in 2021 appears more subdued than all other years observed. The Bear River drainage basin is located in between the Great Salt Lake and Yellowstone National Park in the Rocky Mountain region of the United States. The eponymously named river flows in a counterclockwise loop.

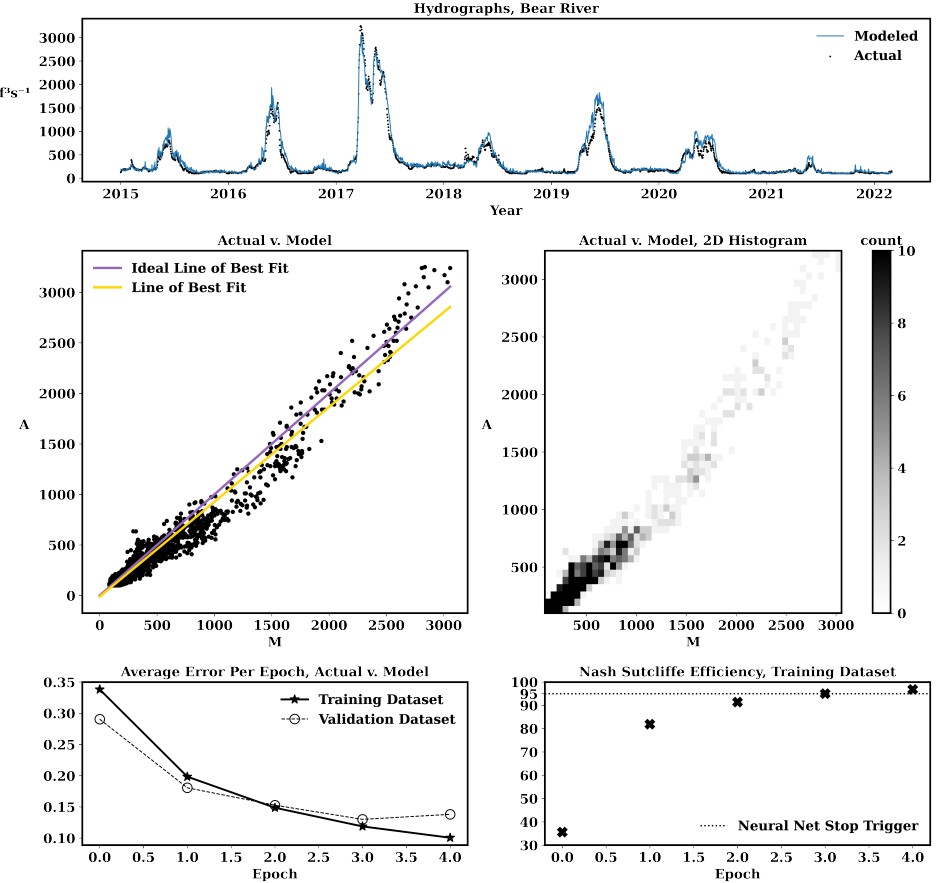

Figure 4: Neural network sample output

The second row plots each modeled observation in the time series against its respective actual measurement. On the left is a study of the model output ordered on the x-axis from low to high flows and corresponding actual measurement on the y-axis. The right plot retains the same axis labels, but instead observes spatial proximity of values. Darker points are more commonly occurring ranges of flow. The left plot also contains two lines of best fit, the ideal or desired line found from the data, and the actual line of fit as exists between the actual gauged streamflow and the neural network model output of streamflow from surface and subsurface flow.

The third and final row shows epochal values during the neural network training process. On the left, the average error between the actual measurements and DCRRNN output declines as the model goes through its iterations of propagate and backpropagate. Concomitant with error v. epoch is efficiency v. epoch. As the error declines towards zero, the NSE measurement increases towards 100%. When

the neural network is set to run, the lever of NSE stoppage trigger is iterable. Here it is set to stop the network at an NSE value of 95%, which occurs at the sixth epoch.

We perform nine iterations of the configuration of 252 experiments. For each of the four basins, there are sixty three experiments per iteration based on nine possible values of lag and seven possible values of data split, equating to 2,268 individual runs of the same neural network. Each experiment either stops when the measurement of average NSE of the training dataset within an epoch equals 95% (see bottom right plot in Figure 4) or the total number of epochs of back and forward propagation of the entire basin dataset reaches 100. The resultant 2,268 database entries include an specialized output within the framework of Figure 4. We execute the experiments through a slurm controlled high performance computer cluster. Computations are constrained to a single node with two central processing units, a single NVIDIA GeForce RTX 2080 Ti graphical processing unit, and no more than 130 gigabytes of random access memory. Our platform is written in the python programming language and managed with the miniconda package manager. The total run time to compute the experiments within was 83.0 hours.

## 4 DISCUSSION

The results presented indicate relatively favorable performance of the neural network architecture as it is applies to the transformation of surface and subsurface flow into a prediction of basin gauged streamflow; the kernel density estimates (KDE) in Figure 5 illustrate this point. We executed a total of more than 2,200 experiments in total using the common architecture. We use two hydrological metrics: Kling Gupta (KGE) and Nash Sutcliffe (NSE) (Nash & Sutcliffe, 1970; Gupta et al., 2009; Gupta & Kling, 2011; Knoben et al., 2019). For each of these metrics, the peak resultant merit value of the 2,268 experiments is greater than ninety percent with a standard deviation of less than 0.06. The results are tolerant to lagging the data beyond the residence time of water in the atmosphere (Van Der Ent & Tuinenburg, 2017; Gimeno et al., 2021).

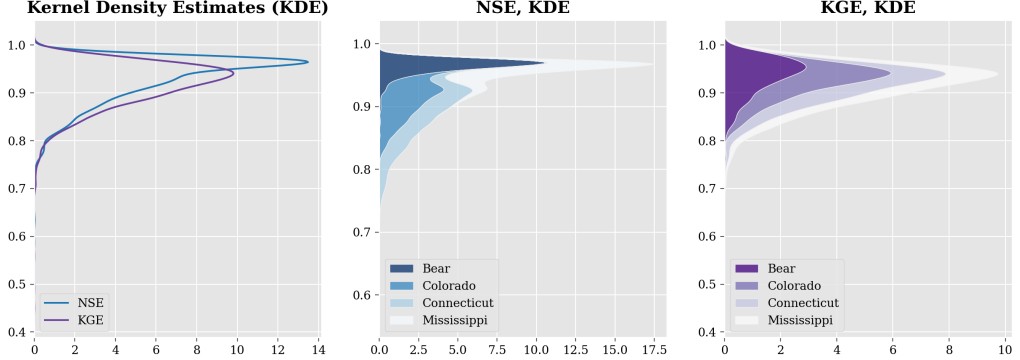

Figure 5: Kernel Density Estimates (KDE) of the 2,268 experiments.

Others have observed the changing water quantity of the Mississippi. One study used NLDAS data focused on a subsection of the Mississippi with a higher quantity of streamflow target sites (Qi et al., 2019). Another group considers a different data system altogether for watershed modeling on the upper Mississippi basin (Chen et al., 2021). Some groups suggest that NLDAS is too simplified, and are decided to create their own blend. They take a much broader approach than the scope of the problem observed here (Tran et al., 2022). The same is true for another study, where they look at several different models and 961 small river basins. There appears to be some disparities in the upper midwestern United States model (Cai et al., 2014). Some use meteorological data as a predictor for electric outages, as seen in a study looking at Connecticut. They, too, use the Nash Sutcliffe efficiency as a figure of merit (Yang et al., 2021) but are approaching the problem with a different lens. Their target is a smaller population and the risk of being without electric power.

This process can be expanded in different ways. Our study relies on the internal programming of NLDAS to compute surface and subsurface flow. There is much uncertainty in these observations based on the natural heterogeneity of the land surface. We do not look at the independent influence

of any single forcing variable. Take snow, for example. In large mountain proximal basins such as those near the Rocky Mountains or Himalayan ranges, accumulation of subzero degrees Celsius water in solid form provides a continuous upland buffer tank for the communities with which the river down land serves. As the relative presence of carbon dioxide increases and the land temperature responds in agreement, the duration and scale of snow melt and sea ice is variable. It is challenging to equate with exact certainty how much solid water exists. To a degree, interpolating satellite data with gauged data is sufficient, but these apparatus are challenging to maintain in cold temperatures or in places of very high altitude. One could elect to observe more individual locations as targets, therefore making the relationship no longer image to single value at a given time, but instead image to image. There are studies that consider the impact of slow moving oceanic and atmospheric abnormalities upon the hydrology of the land. Independent variables include the Madden-Julian oscillation (Jiang et al., 2020), the El Niño–Southern Oscillation (Hu et al., 2015), and the Atlantic meridional overturning circulation (Ionita et al., 2022).

While the NLDAS product used here is of a particular spatial fidelity, the Global Land Data Assimilation System is more coarse in its resolution. It is beneficial to the scientific community to have a clearer picture of the meteorological forcing and environmental responses in the ocean, land, air, and mixed interfaces. One could use this framework to fuse the high resolution NLDAS product with the global GLDAS product and evaluate the result according to one common set of metrics. The software could be packaged and ported to use with an already existent embedded *in situ* mesh system to provide forecasting information.

Instead, one might consider looking at a different time signature, such as seasonally decomposed but over several years or introducing higher resolution localized water quality data into the model. By tracking environmental statistical anomalies relative to other points in time and relative to the global community, municipal decision makers can clue into the trajectory of their land, their structures, and their constituents within. The choice to retreat is not to be approached lightly, but in some instances is becoming the necessary measure (Siders, 2019; Hino et al., 2017). This intelligence can also be placed in the hands of consulting engineers to distribute in new and existing infrastructure. Logic is necessary to manage assets of complex hydraulic systems (pumps, motors, chemical feed, aeration, dewatering, gates, valves).

Lessons must be learned from events on both sides of the water quantity spectrum such as the 2022 Pakistan and Mississippi floods on one end and the 2017 Cape Town South Africa water crisis on the other. The opportunities to improve our monitoring systems are many; however, more people are needed in the conversation to consider how we might better cooperate with the environment.

## 5 CONCLUSION

Using modern techniques and data systems, we introduce a fresh perspective to studying and understanding the water cycle with a learned representation. Our results show that a deep convolutional residual regressive neural network combined with water flux and gauged streamflow data comes to an optimized state, exhibiting strong forecasting performance according to standard hydrological statistical figures of merit. Through the careful use of visuals and data management, this solution is poised to approach with success other locations, degrees of fidelity, time scales, and parameters of interest in the greater climate observatory community.

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
