# OpenReview forum: "Discerning Hydroclimatic Behavior with a Deep Convolutional Residual Regressive Neural Network"
_ICLR.cc/2023/Conference — Submitted to ICLR 2023_

### Official Review · Reviewer_Ck2P · 2022-10-23

**Confidence:** 4
**Correctness:** 3
**Technical Novelty And Significance:** 1
**Empirical Novelty And Significance:** 2
**Recommendation:** 3

**Clarity, Quality, Novelty And Reproducibility:**

The paper reads well in general, but there miss too many key information, such as the literature, the architecture, an explanation on the architecture design, ...
The paper is not novel in terms of methods.
I don't think that the codes are available.


**Strength And Weaknesses:**

Strenghs:
- the application is of importance
- the input data, which comes from real measurement, is of large size and a specific reduction was performed to treat it


Weaknesses:
- the paper does not present in the introduction a state-of-the-art in modelling (data driven and not data-driven) of watershed gauged streamflow, nor a literature on water levels estimation (such as rivers) from machine learning.
- the paper does not compare its results with other techniques, or at least a baseline
- I am not certain how the splitting between train and test was made; it should be clarified. If the splitting is done randomly, there is an over fitting problem: nearby days might have the same behavior. We would need to separate the data of different years, in order to see if we are able to make the model work in the future.
- the machine learning model is not clearly explained, nor how it was chosen. Moreover, I think the technique does not show a novelty which would make it appropriate for the ICLR community.

Typos and small remarks:
-  'ongoing western United States drought' --> not clear, will depend on when the reader reads it, please clarify the dates
- 'Noah': please add a citation
- 'the network that has random numbers associated with each layer and node of the network' --> no, the weights are optimized based on the data so they are not random after training.
- 'a variant of stochastic gradient descent' --> please indicate which one
- 6 epochs is usually too small. This means that you should probably reduce your learning rate.
- 'it is applies'
- 'One study used NLDAS data focused'

**Summary Of The Paper:**

This paper presents a method for estimating the gauged stream flow of a watershed from the water flux daily observations at each location of a river basin. A convolutional neural network is used to perform this regression, and it is tested on real data (4 USA basin rivers).


**Summary Of The Review:**

While the problem is of interest and the data seems to be interesting, this paper is not a good fit for ICLR. Moreover, the technique is very simple, the state-of-the-art and comparison are lacking, and I am not sure about the experiments (there might be an overfitting). I would suggest to clarify this point and submit to a more water cycle applied conference or journal.

---

### Official Review · Reviewer_izoj · 2022-10-25

**Confidence:** 5
**Clarity, Quality, Novelty And Reproducibility:** 1.	Overall, this article does not pro…
**Correctness:** 2
**Technical Novelty And Significance:** 1
**Empirical Novelty And Significance:** 1
**Recommendation:** 3

**Strength And Weaknesses:**

Strength:
1.	Clearly specify the background and problem definition.
2.	Give a detailed description of the dataset, and selected basins used in their experiment, also clearly specify climate driver (from NLDAS) as input and streamflow observation (from USGS) as output.
3.	The author also uses clearly predictive performance to justify the effectiveness of the proposed DCRRNN model.

Weaknesses:
1.	Although the authors mentioned that they proposed a robust DCRRNN model to achieve the goal of accurate streamflow prediction, the detailed model architecture is not explained in the article. It should be clarified in the methodology section.
2.	The authors only use a bunch of quantitative analysis to show the effectiveness of the proposed model, but they did not compare their model with a state-of-the-art method such as EA-LSTM, PGRNN  and etc.
3.	The authors didn’t show the experiment results under limited observation data to further justify the robustness of the proposed model.


**Summary Of The Paper:**

In this paper, the authors proposed a robust DCRRNN model to predict streamflow dynamics by using climate drivers (e.g. air temperature, wind speed and etc. ). The experimental results already demonstrate the effectiveness of the proposed model, which can capture the dynamic change of streamflow from 2015 to 2022. Meanwhile, the authors also show that their proposed model can be applied to different basins under different climate conditions.

**Summary Of The Review:**

Overall, this article does not propose a novel approach to implementing streamflow prediction. At the same time, their experiments have not been enough to prove the effectiveness of the model.  Here are some suggestions:
•	Currently, their model only test on 8 years of data. They should use longer sequence data in experiments.
•	They should compare the performance of the proposed model with other popular methods such as EA-LSTM, PGRNN and etc.
•	They should design different experiment settings to test the robustness of the proposed model.

---

### Official Review · Reviewer_UhiG · 2022-11-01

**Confidence:** 2
**Correctness:** 3
**Technical Novelty And Significance:** 1
**Empirical Novelty And Significance:** Not applicable
**Recommendation:** 1

**Clarity, Quality, Novelty And Reproducibility:**

Clarity: Good.
Quality: Poor
Novelty: Poor
Reproducibility: Poor

**Strength And Weaknesses:**

Strengths:
1. this paper studies an important real-world problem, which is interesting to me.
2. the visualization is easy to follow.

Weakness:
1. the main contribution is very limited. In my opinion, this paper combines several functions together to make a tool for the application. As a matter of fact, I don't find something new or something useful. The proposed method DCRRNN is mainly in Section 2.5. Is it really a novel ML model? It only includes some implementation details, which cannot be called as "method" from a research paper.
2. the whole paper is a simple application, which cannot meet the basic requirement of ICLR for a research paper.

**Summary Of The Paper:**

This paper mainly studies the basin response of watersheds to water cycle fluxes. For this aim, this paper proposes a platform to visualize, analyze and conduct forecasting. Experiments show its effectiveness in simulating and forecasting the impacts of hydroclimatic events.

**Summary Of The Review:**

The overall contribution is very weak. The quality is limited. It's not a proper paper for ICLR.

---

### Decision · Program_Chairs · 2023-01-20

**Decision:**

Reject

**Justification For Why Not Higher Score:**

This paper has methodological flaws in the evaluation that require extensive re-analysis and rewriting.

**Justification For Why Not Lower Score:**

N/A

**Metareview: Summary, Strengths And Weaknesses:**

This paper presents the use of a conv net architecture for help in forecasting hydroclimatic events from satellite images of watershed basins. The study is conducted over 4 watersheds in the US using images of water flux (surface and subsurface runoff) from the US National Land Data Assimilation System, as well as ground truth river streamflow measurements used as targets for the predictor. The paper discusses limitations of the data and specific factors that can influence the predictions.

All reviewers agree on the importance of the problem, and also praise the description of the dataset and of the metric (izoj).

Reviewers note that this is a simple application paper, using a standard conv net (UhiG), find the evaluation insufficient w.r.t. state-of-the-art or even w.r.t. a baseline (izoj, Ck2P), and find that the robustness of the method is not well evaluated for limited observation data (izoj) or for train-test split (Ck2P). There are also issues of clarity (izoj, Ck2P).

Given scores of 1, 3, 3, this paper does not meet the acceptance bar for ICLR. I would suggest rewriting this paper for clarity, following the reviewer suggestions for better robustness analysis, and resubmitting it to a climate workshop.

**Summary Of Ac-Reviewer Meeting:**

N/A